# Exploring Islamic Spiritual Care: What Is in a Name?

**Naveed Baig** [1,*] and **Nazila Isgandarova** [2]

1  Faculty of Theology, University of Oslo, 0371 Oslo, Norway
2  Emmanuel College, University of Toronto, Toronto, ON M5S 1K7, Canada; nazila.isgandarova@utoronto.ca
*  Correspondence: naveed.baig@teologi.uio.no

**Abstract:** At present, there is limited theoretical clarity on the nature of Islamic spiritual care, which is a developing discipline around the world. How it is defined will be instrumental for spiritual care institutions, professionals and recipients of spiritual care in the years to come. This article wishes to understand and explore the idea and vision behind Islamic spiritual care and why this line of investigation may be of importance to care providers with different faith backgrounds.

**Keywords:** spirit; spiritual care; Islamic spiritual care; Islam; healthcare; psychotherapy





## 1. Introduction

Islamic teachings pay close attention to the spiritual aspect of health and encourage its followers to achieve balance between body, mind and soul in order to enjoy sound health. The word "Islam" refers not only to submission to the Divine but also to peace-making, wholeness and well-being (Reda 2012; Ghazi Bin Muhammad 2017).

The Qur'an and the prophetic tradition extensively address spiritual issues and concerns because they affect the health of the person in a variety of ways. Therefore, Islamic teachings value and consider spiritual care as primary 'care' in private, social and institutional settings. Historically, Muslim healthcare practitioners and spiritual caregivers not only diagnosed and treated physical illnesses but also provided psychosocial and spiritual intervention to address the spiritual dimensions of illness (Gilliat-Ray et al. 2014; Keshavarzi et al. 2020; York Al-Karam 2018; Rassool 2019; Isgandarova 2018).

In Islam, the whole person is viewed from three aspects: biological/physical, psychological/emotional and spiritual. Islamic theology maintains that the spirit (*ruh*) is the core of the being, which humans pursue as the perfection of being. Although there is still a gap in the contemporary healthcare system to address spiritual aspects of health (i.e., lack of funding for spiritual care in institutional settings, the medical model of healthcare vs. social care of healthcare still fails to see the role of the spiritual heart (*qalb*) in mental health (Rothman and Coyle 2018)), spirituality is becoming popular in health care. Healthcare practitioners are becoming more aware that spiritual distress may occur "when the person does not find meaning, hope, love, peace, comfort, on connections in life, or when they experience a conflict between what they believe and what actually happens in their lives" (Levitt 2005, p. 63).

This article aims to elucidate the field of Islamic spiritual care and explain why this inquiry may be important to care providers with non-Muslim backgrounds. After providing a general definition of spirit and spirituality, we will provide insights into the theological and philosophical paradigms of the Islamic tradition in relation to spiritual care and discuss why this knowledge may be of use to care providers who are not Muslim. This article intends to answer the question: How can Islamic spiritual care be understood today, and what use may it have for care providers of other faiths or no faith?

## 2. Methodology and Method

### 2.1. The Authors' Stand

The two authors of this article have many years of teaching and clinical experience working with patients and relatives, as well as hospital staff from a range of different religious and ethnic backgrounds. Even though the clinical settings in which we work (and our approaches)—Canada and Denmark, respectively—are different, there is a consensus on the textual sources of Islamic spirituality. The central sources of Islam are derived from the Qur'an and Sunnah (Long and Ansari 2018, p. 110). This creates a generic understanding of Islamic spirituality that transcends the different fields of care, from personal to collective, including hospital and prison chaplaincy, marital and drug abuse, counseling and caring for family, neighbors, congregations, etc. This article is primarily concerned with Islamic spiritual care in healthcare, but its content can also be useful for readers who have an interest in Islamic spiritual care in general.

The authors' professional approaches, the psycho-spiritual and counseling approach, on the one hand (N.I.), and the theological and pastoral care on the other (N.B.), are areas of exchange, discussion and even tension. In this article, we tried to use and discuss voices from different fields of social studies, like sociology, theology and psychology, to give a theoretical overview of our understanding of Islamic spiritual care. For this, we gathered relevant texts using search engines and databases, such as ATLA Religion Database with ATLASerials (EBSCOhost databases), PubMed and PsycINFO. There are practices, understandings and underlying values from all of these fields that can be helpful for people needing care. According to the authors of this article, theology and psychology have long and integrated historical roots, and their higher objectives are not mutually exclusive but, at best, complement and strengthen each other. Our contention is that there is always a worldview behind an action that can be rooted in religion, rationality, sentimentality, political ideologies, etc. In other words, behind every action, there is an 'authority at 'work'', sometimes consciously and sometimes unconsciously, that is a part of who we are and the lens through which we see the world. Which "authority" ensures healing or curing at hospitals? Bluntly said, is it a Higher Power/God, healthcare system or both that cure the patient? Can the use of religion and spirituality cure patients from disease and suffering? Have science and medical advancement in recent times not shown us that it has the answers for alleviating pain and curing disease? These and other pertinent questions are part of ongoing discussions where borders are being drawn and redrawn to accommodate the different authorities at public institutions like hospitals. For example, in Denmark, Christian pastors are sitting on state-run hospital ethical committees and palliative care inter-disciplinary teams. In Canada, Muslim psychotherapists and spiritual caregivers provide spiritual support for patients with different beliefs and life stances.

### 2.2. Method

Islamic spiritual care belongs to the field of Islamic practical theology. In Islamic academic circles, practical theology, in comparison to Christian theology, is a relatively new discipline (Isgandarova 2016; Abu-Shamsieh 2019). In Christian practical theology, various models of 'mutual critical correlation' have been used primarily by practical theologians, where hermeneutics, co-relationality and theology are at center stage, interacting with each other critically (Miller-McLemore 2011; Swinton and Mowat 2016). Also of importance in these models is the usage of experience (product of rational reflection) and reason, whilst also acknowledging the fact that theology needs to enter into a deeper dialogue with other sources of knowledge, such as the social sciences (Swinton and Mowat 2016). As hermeneutics of Islamic spiritual care, practical theology can play a pivotal role in creating correlations between: (a) religious/spiritual practice and its tradition and sources and (b) between theological and social-scientific accounts of religious/spiritual practice. The authors' method in this article is theological inquiry and reflection. We correlate our many years of spiritual care practice with Islamic sources and engage with social sciences in exploring Islamic spiritual care.

### 3. Theology and Psychology

Regarding the relationship of spirituality to psychology, we would like to recognize the work of the deceased Sudanese Muslim psychologist Malik Badri (d. 2021), who, since the 1960s, expressed his disapproval of some parts of modern psychology. He stated that modern psychology is "influenced by Western secularism and its ungodly worldview and its deviant conceptions about the nature of man" (Badri 2009). Badri intends to give modern psychology an Islamic expression, and his blatant confrontation is an example of the friction between religion and modern psychology.

Badri's position has also been observed among non-Muslim healthcare practitioners. For example, a study from a Danish Ph.D. dissertation has shown that Danish chaplains (from the majority Lutheran Church) deal with complex linking and hybridization of religious and therapeutic discourses (Raakjær 2019). The 'therapeutic ethos' of society has not ruled out the role of Evangelical Lutheran Christianity or transformed the hospital chaplains into semi-therapists (Raakjær 2019). On the contrary, the therapeutic culture strengthened the chaplaincy profile (Raakjær 2019). However, the research finds that chaplains use and practice tools of psychology to serve their own theology (O'Connor et al. 2014; Isgandarova 2018; Cadge and Rambo 2022). In the last few decades, psychologists have also used spirituality and religion to understand the existential meaning assigned to suffering (Pargament 2007).

In both cases, Badri and Raakjær (with a Muslim psychologist and a Danish Christian pastor background, respectively) seem concerned about the 'authenticity' of their respective traditions being influenced by 'external infringement'. Badri is highly critical of Muslim psychologists who blindly follow Western psychological thought (source) and the Danish title of Raakjær's Ph.D. dissertation, *'Fra frelse til well-being?'* (From salvation to well-being?) highlights very well the perceived schisma, and maybe even fear, between the fields of religion and modern therapeutical culture.

### 4. Definition of Spirituality

Defining spirituality seems to be a "mammoth" task as there is a lack of conceptual clarity in this field. "Currently, there is no 'gold standard' for the definition of spirituality that can be established independent of the historical use of the term in the English language or the Greek or Latin roots from which the word 'spirituality' is derived" (Reinert and Koenig 2013, p. 63). According to McSherry (2006), the word 'spirit' originates from the word 'spiritus', "which generates images of life, breath, wind and air" (p. 45). The two authors outline four keywords that are commonly associated with 'spirit': spiritual needs, spiritual well-being, spiritual distress and spirituality. The 'spirit' "drives and motivates individuals to find meaning and purpose, allowing expression in all aspects and experiences of life, especially in times of crisis and need" (McSherry 2006). This definition of the spirit can be defined as an individual entity of its own, capable of 'moving' humans and allowing them expressions, especially during times of suffering.

In general, "spirituality as meaning and purpose is seen to manifest itself in a quest towards self-actualization, and the search for human integrity"; spirituality and connections and relationships refer to an "individual's relationship with self, other, the cosmos, and god/God"; a relationship with God/Transcendent One is the origin of spirituality; spirituality "as Transcendent Self relates to the experience of the transcending the self so as to identify with the experience of another person"; spirituality as a vital principle means that it "vitalizes the whole person and/or the cosmos"; as a unifying force/integrative energy, spirituality is a "non-personified, incorporeal common energy that unifies reality"; etc. (McCarroll et al. 2005, pp. 45–47). Regardless of how one experiences spirituality, it is defined and experienced in five ways: "spirituality is known practically; spirituality is known phenomenologically; spirituality is known linguistically; spirituality is known in subjective experience; spirituality is an unknowable mystery" (McCarroll et al. 2005, p. 49). Further, it reflects the "distinction between the realm of the eternal (the good, universal, uncreated, unchanging) and that of the historical (particular, created, changing), in which the latter is relativized and limited by the former" (McCarroll et al. 2005, p. 55).

Apart from the etymological understanding of spirituality mentioned above, Aarvik (2020), a Norwegian sociologist of religions, argues for a phenomenon she calls 'spiritualized Islam', drawing upon her research from young non-organized Muslims in the West. Islam epitomizes a "synthesis between spirituality and religion in the way in which it combines and sacralizes both subjective life and objective elements of Islamic dogma", according to Aarvik (2020, p. 94). Thus, this type of spirituality, which challenges the boundaries of Islamic orthodoxy and simultaneously calls for self-realization and a connection to God, eliminates the religion–spirituality dualism. Aarvik's research suggests that the different interpretations of Islam form a synthesis of two different approaches to the sacred: 'life as religion' and 'subjective-life spirituality', as described in the work of Heelas and Woodhead (2005).

In Islamic tradition, any discussion of spirituality is directly linked to the notion of Islam as a 'discursive tradition'. Talal Asad (2009) defines a discursive tradition as a discourse that seeks to guide practitioners about a given practice's correct form and purpose. Further, it seeks conceptually to link a past and a future through a present. This allows Muslim spiritual caregivers to transcend literal and fixed interpretations of key Islamic sources, including the Qur'an, to allow a time-free approach to Islamic spiritual care. Nevertheless, any "conceptions of the Islamic past and future" should be linked to "particular Islamic practice in the present" (Asad 2009, p. 20). This 'discursive tradition' is grounded in the classical past and enrobes one with practices and ideals for cultivating the now and envisioning the future. Hence, Asad's 'discursive tradition' can open gateways for a type of spiritual care that is rooted in tradition and, at the same time, meaningful in the present.

## 5. Spiritual Reality—Beyond Religious Formality

Kenneth Pargament (Pargament 1997) and Paul Rennick (Rennick 2005) suggest that spiritual reality is beyond religious formality, and that religious traditions cannot claim that spiritual activity and experience can only be felt in the formal context of religion. Yes, religion can help persons to live the spiritual reality in the communal experience; spirituality can also be lived through individual references and experiences. Nevertheless, it should not be underestimated that spirituality and religion are the 'two sides of a coin' for many. When communal experience and individual experience are severed, both sides run the risk of developing a solipsistic perspective that refers to nothing outside itself. "Although distinctions between religion and spirituality can be made, too much distinction creates distortion" (Rennick 2005, p. 28). Rennick argues that this is the case in contemporary psychology, "which explores and promotes a notion of spirituality that is severed from religion and dismissive of theology" (Rennick 2005, p. 29). He explains this with the notion that:

> "theology and psychology have been seen as competing systems of meaning rather than as complementary ones. Spirituality becomes an acceptable category for psychology, since, when it is severed from religion it can be voided of its sacred meanings and added to the psychological lexicon". (Rennick 2005, p. 29)

In general, the definition of spirituality usually includes themes such as "meaning and purpose; connection and relationships; God/god(s); transcendent Other; transcendent self; vital principle; unifying force or integrative energy; personal and private; and hope" (McCarroll et al. 2005, p. 45). Religion is defined and explained within the framework of participation in organized religious institutions and formal engagement in religious activities such as individual or collective prayers. However, the themes of both spirituality and religion are directly related to human needs and experiences. In this respect, they can both inform and enrich psychological disciplines. Moreover, we should also not forget the religious roots of psychology, i.e., *psyche* (soul) and *logy* (study), that became a profession in the 19th century and was subjected to scientific inquiry. Nevertheless, as Pargament (2007) notes, regardless of differences, all the major paradigms of religion, spirituality and psychology are embedded in human experience and "share an interest in helping people maximize the control they have in their lives" (p. 11).

## 6. Spirituality—Meaning and Purpose

The theme of meaning and purpose in spiritual care is "a result of the spiritual experience or spirituality, rather than the essence of the spiritual experience or spirituality" (Reinert and Koenig 2013). Such an approach might disregard the role of the process of meaning-making during spiritual care, but it is still helpful to relate spirituality to meaning and purpose as a dominant theme.

The various definitions and approaches toward spirituality also suggest that spirituality is beyond human comprehension and cannot be controlled. Therefore, it provokes challenges to research and clinical practice. Nevertheless, diversity in definitions of spirituality constitutes the richness of spiritual experiences. Therefore, such richness and diversity in the name of spirituality reflect individual and subjective experience within the context of belonging to the greater. As multiple studies pointed out, many people use their spirituality and religious beliefs to support and guide them in times of stress. It is a source of coping with significant life events and daily life problems. In some cases, a belief in a Higher Power is the answer to their question, "Why me?" and some patients believe that "God had a reason" (Pargament 2007, p. 10). In many cases, spiritual resources are critical in the recovery process. In sum, as theologian Paul Tillich (1952) notes, questions related to "ultimate anxiety", which refers to "the anxiety of fate and death, the anxiety of emptiness and meaninglessness, and the anxiety of guilt and condemnation. These deep questions seem to call for a spiritual response" (Pargament 2007, p. 11).

## 7. The Mapping of the Human Being in Islamic Spirituality

In Islam, the word spirit or breath of life (*ruh*) has its foundations in the Qur'an, where the root of the term is mentioned 19 times (Osama 2005). The term *ruh* is used in the Qur'an to refer to various metaphysical entities like angels, Archangel Gabriel, revelation and divine inspiration, although it often connotes the most inner essence of humans (Baig 2022). Even though the Qur'an offers scant information about the *ruh*, it does, nevertheless, mention humankind being gifted a unique position in the universe containing a 'Divine spark': "When I have fashioned him (Adam) and breathed into him of My spirit, fall you down in prostration unto him" (Qur'an, 15:28–29). In general, the spirit has been interpreted as "the life-source and a subtle substance that makes each being unique and original" (Isgandarova 2018).

Although the spirit is essential in mapping the essence of humans, based on the Qur'anic challenge, all Muslim scholars unanimously agree that human beings cannot comprehend certain realities regarding the spirit (Elahi 2007). Such an argument comes from "human beings' inability (to comprehend or explain) certain things that happen to themselves", such as the extraordinary and supranatural events, miracles and other matters related to the unseen worlds (Elahi 2007, p. 30).

Other elements that constitute the human being are also indispensable in determining the faculties and functioning of the human being. These other elements are the heart (*qalb*), intellect (*aql*), soul or self (*nafs*) (Isgandarova 2019; Keshavarzi et al. 2020; Haque 2004). The composition of the human being has many names because of its accidental modes or states (*ahwāl*). Al-Attas (1995) suggests that when the human is:

> "involved in intellection and apprehension it is called' intellect'; when it governs the body it is called 'soul'; when it is engaged in receiving intuitive illumination it is called 'heart'; and when it reverts to its own world of abstract entities it is called 'spirit'. Indeed, it is in reality always engaged in manifesting itself in all its states". (p. 148)

Hence, these components of the human are one entity, indivisible and unified. Also included in these states is the "*nafs*", which is often translated as an 'ego' or 'animal self ', which can fall down "to the lowest foothills of the bestial nature" (Al-Attas 1995, p. 147). On the contrary, faith, good morals and sincerity can inspire humans to soar toward the angelic realm where divine peace thrives (Al-Attas 1995, p. 146).

The physical body (*djism*), its impact and its importance are often rare topics of discussion around spirituality in Islamic literature. Nevertheless, it is an important theme

since the human being is not only created by God but is shaped 'in God's form'—with a purpose—and the holistic worldview of the human being in Islam, therefore, obliges one to perceive the physical body as a part of the grand scheme of things. For Al-Ghazali, the spiritual journey toward God needs the body, as he himself states in allegorical terms:

> "...the body may be figured as a kingdom, the soul as its king, and the different senses and faculties as constituting an army. Reason may be called the vizier, or prime minister, passion the revenue collector, and anger the police officer". (Al-Ghazzali 2016, p. 22)

Al-Ghazali (d. 1111 AD), an Islamic scholar, mystic and philosopher who was interested in creedal matters, dogmatics and systematic theology, proposed an anthropocentric approach, which is another way of engaging with people in tribulations by taking care of the whole person. He, for example, introduced the concept of 'self-realization' where, according to him, humans must discover their 'selves', their ultimate purpose and the reasons behind their misery and happiness (Haque 2004). Ghazali's own spiritual journey, leaving his hometown, family and influential teaching position at an Islamic seminary, was a voyage from a theistic and theoretical worldview toward an empiricist and person-centered understanding of the world and God.

Thus, according to Al-Ghazali, the revenue collector and the police officer must be subservient to the king, but if they rebel and fight against the king, the soul (king) and, thereby, the body (kingdom) will be ruined (Al-Ghazzali 2016).

## 8. Spirituality and Sufi Cosmology

Islamic spirituality is often correlated with Sufism, the spiritual tradition of Islam, which is often called *tasawwuf* or Islamic mysticism (Nasr 2007). Therefore, it would be helpful to outline the Sufi interpretation of spirituality, which is heavily linked to doing 'spiritual acts' with 'correct intentions'. In Sufism, *doing beautiful acts* (*ihsān*), servitude and charity (*khidmah*), good morals and character (*husn al khulq*) are all interlinked and fall under the umbrella of *ruhāniyah* (Islamic spirituality). Traditionally, in Sufism, spirituality was defined in the context of religious experience and a relationship with God, who was seen as Supreme, Transcendent, All-Powerful, Merciful and Compassionate. However, there were several periods in Islamic history where spirituality transcended formal religion and gave new meaning and impetus to it. In Sufi cosmology and its lived spiritual experiences, it was not only accepted but was a primary goal for an individual to strive for an intimate relationship with the Prophet Muhammad and God. One strand of Sufism focused on intense love (*ishq*) for God, also known as the 'Sufi love theory' (Lumbard 2007). For the Sufis, this is a voyage of discovery, a type of love affair with the divine beloved in which the lovers merge in mystical union, where love is the beginning, the means and the end. Love is the raison d'être—so much so that Attar, the Persian Sufi poet, is said to have uttered: *lā ilāha illā 'ishq*—there is no God but love (Lumbard 2007). These experiences of the Sufis were subsequently written, expounded, theorized, and debated. A considerable amount of Sufi shaykhs, their disciples and others belonging to the Sufi order (*tariqah*) have transmitted extensively, from treatises to poetry, novels to eulogies, revealing their spiritual states and inner thoughts (Ernst 1997; Nasr 2007). All of them encouraged Muslims to search for the truth, either through personal religious or spiritual experiences (*ma 'rifa*) or from the knowledge (*'ilm*), which they may acquire through books or through inspirations (*'ilhām*).

Therefore, Islamic spiritual care practice aims at these aforementioned goals to help human beings achieve their full potential. For this purpose, any terms connected to the spirit or spirituality have an immense significance in Islamic spiritual care concerning assessments, diagnosis and treatments in clinical settings. This is because it directly relates to spiritual qualities such as will (*irādah*), steadfastness (*istiqāmah*), freedom (*hūriyyah*), etc., in reference to the spiritual health of the person.

## 9. Islamic Spiritual Care in Contemporary Times

Islamic spiritual care in Islam is not new, but the professionalization and organization of it in the form of Muslim chaplaincy is (Long and Ansari 2018). Since the emergence of religious traditions in history, spiritual caregivers have been active in the provision of spiritual care to those who need it. In the Christian context, they were called either ordained or non-ordained ministers or chaplains and in the Islamic context, they were called imams or shaykhs. The word chaplain (or *capellanus* from *capella* or chapel in French and *chapelain* in Old French) described the clergy who provided spiritual and religious work outside the formal religious institutions. According to the Oxford Dictionary of the Christian Church and Webster's New Universal Unabridged Dictionary, they "were the capellanus, or keepers of the cape". In many instances, faith groups or organizations commissioned their clergy or laypersons in an institution, organization or government entity (Hunter 2005). The main duty of these "walking" ministers was to provide spiritual care to people who were not able to attend their faith institutions.

In the monotheistic religions of Judaism, Christianity and Islam, the notion of 'Abrahamic shepherding' is reminiscent of pastoral care that originates from the Latin word meaning "shepherd". This type of care, or caring for the 'flock', is a metaphor for family, neighbors and human beings' friend or foe (Ansari 2022). According to Ansari (2022), there is no contradiction between the Judeo-Christian understanding of pastoral care and the Islamic view of its practice. (Ansari 2022). "God did not send any Prophet but that he cared for sheep" and "every one of you is a shepherd and responsible for his flock" are sayings attributed to Prophet Muhammad, examples of theological textual evidence suggesting caring as a responsibility (Al-Bukhari n.d.a, p. 2262).

Islamic spiritual care is based on the concept that humans are a composite of integral physiological, psychological, mental and spiritual components. Muslims seek early medical attention, according to the Prophet's practice and teaching, because a healthy body and spirit is a gift and trust from God. However, the Prophet's definition of *ihsān*—doing what is beautiful—sets out the criteria for Islamic spiritual care and points toward vigilance and the highest level of self-awareness, including professional awareness in Islamic spiritual care (Isgandarova 2012).

Islamic spiritual care is sometimes defined in the context of denominative affiliation and identity. Mohamed Ahaouaou suggests that "spiritual caregivers from the various religious and worldviews are not interchangeable" (Ajouaou 2014, p. 42). Such an approach to spiritual care can be limited and does not leave a space for universal aspects of spirituality. Ajouaou sees spiritual care for the whole person in terms of religious needs. Nevertheless, spiritual needs, along with religious needs and concerns, have also been at the heart of Islamic spiritual care since its inception. Furthermore, conceptualizations of Islamic spiritual care depend on personal preferences. For example, one's cultural and ethnic background may influence one's approach to Islamic spiritual care: it can either be spiritual or gnostic, religious or secular in nature. A spiritual care provider such as a chaplain is expected to see beyond differences of faith and opinion and try their best to care for all members of their faith community and beyond (Long and Ansari 2018).

Although there is literature and research on the secularization of chaplaincy in Western institutions, like prisons (Christensen et al. 2020) and hospitals (Stifoss-Hanssen et al. 2019), there is no research on how Islamic chaplaincy and spiritual care in Islam are influenced by the secularity of Muslims living in the West and vice versa. How does a Muslim spiritual care provider engage with the secular, the secularity of Muslim patients and the secularization of institutions in which it operates? Alluding is also the question of how the secular and its contingent terms are understood from a Muslim worldview. Exploring the humanist chaplaincy function may provide fruitful insights to understand chaplaincy in secularized societies. This is because humanist chaplaincy does not fit within the more traditional, religion-focused views of the profession, which has several decades of history in Holland and institutional presence in Northwestern Europe (Schuhmann et al. 2021).

## 10. Islamic Spiritual Care and Its Relationship to Other Health Care Professions

The Prophet Muhammad encouraged Muslims to look for all possible measures to establish health by stating: "God has not created a disease without also creating its cure" (Al-Bukhari n.d.b, p. 5678). A positive attitude toward medical treatment is well expressed by viewing medicine as "a religious vocation of the first order because it helps men and women to help others preserve and restore health" (Rahman 1998). He also encouraged people to feed the hungry, visit the sick and set free the captives, and he himself always prayed and consoled the sick by saying: "No fear, it [illness] is a catharsis, God willing" (Rahman 1998). The Prophet empowered the sick by bringing attention to their spiritual purity that "God hears their prayers" (Rahman 1998). Therefore, the visitation of the sick is a communal enterprise for Muslims and is deeply embedded in Islamic theology.

However, Islamic spiritual care is beyond visiting the sick. It is a more professional obligation and duty. Therefore, since the birth of Islam, Muslim scholars devoted their energy to improving the quality of spiritual care. Both early and modern Muslim jurists considered the science of medicine as an essential science for preserving health and preventing disease. Some of these jurists were also well-versed in the science of medicine (Ghaly 2008, pp. 105–43). Al-Shāfi'i (d. 810), the founder of one of the central Sunni schools of jurisprudence, not known as a medical scholar, said, "Do not live in a place where you have no scholar to inform you about your religion and no physician to inform you about your body" (Ghaly 2008, p. 110).

Al-Dhahabi (d. 1348), a prominent Muslim scholar and historian, defined spiritual care in light of the benefits of Islamic ritual prayers that contribute to the client's spiritual, psychological, physical and moral well-being. For him, effective spiritual care with ritual prayer is a form of worship and has a psychological benefit by helping to concentrate on prayers and diverting the mind from pain; it involves certain bodily movements, which cause some organs, such as the muscles, to relax and produces happiness and satisfaction; they suppress anxiety and extinguish anger (Al-Dhahabi 1996, p. 140).

On the other hand, Muslim medical professionals, including doctors, consider it a religious and spiritual duty to educate Muslims with respect to important aspects of health (Rahman 1998, p. 79). For example, in his introduction to his book *Guide for Students* (*Hidāyat al-Muta 'allimīn*), Abu Bakr Rabi' ibn Ahmad al-Akhwāni al-Bukhāri (d. 983), the student of a famous Muslim physician Abu Bakr Muhammad Abu Zakarīya al-Rāzi (d. 925), stated, "Wise men have said that it is incumbent upon every person to learn [the basics] of the Sacred Law, for when a person knows the Sacred Law, he is immune from going astray. Second, he must know some [basic] medicine to preserve his health so that quack doctors will not be able to destroy him. Third, he must learn some art to earn his livelihood by lawful means." (Rahman 1998, p. 39).

In Islamic tradition, al-Rāzi (d. 925), al-Kindi (d. 873), Ibn Sina, or Avicenna (d. 1037), Shams-ul- Dīn al Dhahabi (d. 1348), Ibn al-Qayyim al Jawziyyah (d. 1351) and Jalal Ad-Dīn al-Suyūti (d. 1505) are considered to be the most prominent writers of religion and health (Yucel 2007). For example, Abu Zayd al-Balkhi (d. 934) is one of the world's first known cognitive psychologists—he prescribed treatment for anxiety and mood disorders and criticized some doctors for failing to pay attention to spiritual aspects of health (Badri 2013). Al-Balkhi differentiated between neurosis and psychosis and prescribed different techniques to treat these classified disorders. Ibn Sina, who was a famous Muslim surgeon and philosopher, also supported an integrated approach to health care. He suggested spiritual care that starts with prayer and continues with proper medical treatment (Dogan 1997, p. 7).

In the Islamic context, spiritual care is unique in the sense that it dovetails principles and practices from the traditional Islamic sciences together with a psychological understanding of human felicity and growth. This integrative and holistic understanding is from the cradle of Islamic civilization, where Muslim religious scholars were theologians, astronomers, mathematicians, medical practitioners and poets all at once. However, the specialization of sciences and the general development in the West today has changed the scene:

"Within the specialist branches of modern medicine, it can no longer be assumed that Muslim medical professionals are able to provide the holistic approach of their counterparts from earlier centuries. Seen in this light, the evolving role of Muslim chaplains in healthcare today could be seen perhaps as helping to revive the spiritual dimensions of patient care, this time as part of a multi-professional medical team." (Gilliat-Ray et al. 2014, p. 34)

Hence, this leads to the question of how the spiritual care provider sees themself and the institutional guidelines/boundaries. Are the Muslim chaplains or chaplains at large psychotherapists or spiritual counselors? Do they give existential support, solace, hope and meaning alone, or do they diagnose, assess, cure, document and conduct interventions and treatment?

Seeing chaplains as psychotherapists, which is becoming a common picture in Ontario, Canada, where they are often called psycho-spiritual therapists, may be a way of drawing attention to the essential meaning of psychotherapy, which historically was 'care of the spirit'. Therefore, seeing spiritual care as overlapping with some aspects of psychotherapy may help psychotherapists explore the benefits of integrating spiritual resources into the care of their clients. This is an explosive but concurrently timely discussion where there is a persistent calling for more interdisciplinary collaboration between all stakeholders at institutions working to bring light and compassion to the dark and murky rooms of suffering.

## 11. Islamic Spiritual Care: Universal or Specialized Spiritual Care?

The increasingly multicultural and multireligious public care institutions in the West have largely accommodated and developed spiritual care programs, including chaplaincy services for catering to patients' religious, spiritual and existential needs (Gilliat-Ray and Arshad 2016). Hence, how Islamic spiritual care is understood will be critical for its potential acceptance and usage at institutions.

What use can Islamic spiritual care have for care providers with a spiritual or religious belief other than Islam?

1.  Islamic spiritual care gives insight into the outlook on life and a compassionate approach to care in Islamic tradition.
2.  This, in turn, can be used to cater to the spiritual needs of its adherents by care providers of different faiths or no faith.
3.  The worldview and value system of Islamic spiritual care can inspire, nourish and challenge caregivers' own positions and faith traditions.

In this respect, Islamic spiritual care can also be used for care recipients who are not Muslim by being more open in terms of engaging religious differences, listening to and addressing emotional, mental and spiritual distress and making appropriate referrals.

The aforementioned points are based on the foundational Qur'anic principles of love, mercy, compassion and universalism. In this respect, not only the Qur'an but also many Muslim spiritual leaders, i.e., al-Ghazali (d. 1111 AD) and the Andalusian Sufi Muḥyi al-Din Ibn al-Arabi (d. 1240), provided insights for practicing the values of love, mercy and empathy. Contemporary Muslim spiritual care providers, psychotherapists and other mental health providers (Rothman and Coyle 2018; Isgandarova 2019; Long and Ansari 2018; York Al-Karam 2018; Keshavarzi et al. 2020; Ansari 2022 and others) also encourage not only a theistic and theoretical worldview but also a person-centered understanding of the world and God.

As Shaikh (2012) outlined in her work, Ibn 'Arabi also believed that God created the first human in God's own image because of God's deep longing for intimacy. Such a position in Islamic spirituality enables Muslim healthcare providers, including Muslim spiritual caregivers, "not only to become highly tolerant to religious differences but to accept those differences as the client's universal truth" (Isgandarova 2019, p. 101). It also helps them to avoid the pitfalls and dogmatic assertions of organized religious traditions. Isgandarova

also noted that the divine origin of all creation "places love to humanity and empathy for all at the foundation of Muslim healthcare professionals' work" (Isgandarova 2019, p. 104).

*Theistic vs. Anthropocentric Worldview*

Since Islamic theology has a strong—but not exclusive—inclination toward a theocentric worldview, Muslims can find it awkward to talk about spirituality without referring to God (Rassool 2016). The anthropocentric worldview, as Hanafi—a Muslim philosopher—advocates for, does not stand in contrast with theism, but encourages a 'cultural revolution' in pursuit of liberty and social justice (Manshur 2021). These two approaches are important to grasp. By focusing on the individual (in our case, the care service recipient) and how they navigate through suffering, often attempting to make meaning of it, using coping resources from their faith tradition, reflecting on its emotional states and taking stock of them, can be interpreted as an anthropocentric undertaking. The theocentric approach theorizes on God's role and responsibility during one's tribulation—having ties to the theological theme of theodicy—where one reflects on the role of evil in the world. The two approaches need, however, not be in opposition to each other and may even be adopted by care recipients in concomitance.

What Gilliat-Ray in 2015 described as a developing 'Islamic pastoral theology' led by British Imams and scholars working in the chaplaincy field is taking shape in many Western countries today—both in practice and theoretical foundations. In Islamic spiritual care work today, preaching and teaching creedal matters to care recipients is not the rule but an exception. In much of the Islamic spiritual care theology established in Northern America by Muslim chaplains, psychotherapists and others, the focus is very much on care and compassion (Kholaki 2020). For instance, Kholaki has reflected on her work as a female chaplain at hospitals in the US in a chapter on chaplaincy where she presents the five 'whats' of Islamic pastoral care. They include hospitality as a central tenet, which involves creating a welcoming environment, using 'presence' in chaplaincy work, active listening, compassion and connection—regardless of the care-seekers faith background. These five 'whats' of Islamic pastoral care can be ascribed to many of today's faith and non-faith spiritual care traditions. However, 'specialist pastoral care' may include ritual rites and practices that specialists from a particular tradition of faith may best provide. One concrete example is the Qur'an recital during the last stages of life for a Muslim patient or the anointing of the sick in the Catholic tradition.

The book 'Mantle of Mercy' is an example of an 'Islamic pastoral care theology', where Muslim chaplains and activists from different fields elaborate on their understandings of spiritual care from their lived practice and encounters with care recipients (Ali et al. 2022). Put bluntly, the focus is not on how to correct the care recipient's creedal discrepancies but on how a caring practice engages with suffering and tribulations. In short, the Islamic pastoral care theology that has evolved in the last years, particularly from Muslim chaplains working in the West, gives care providers from all faiths (and no faith) an opportunity to assist and shepherd care recipients with Muslim backgrounds, bearing in mind their unique and specialized care needs that may arise.

## 12. Conclusions

As we have seen, it is not an easy task to define spirituality or Islamic spiritual care. According to Islamic sources, the human being consists of many dimensions, including body and spirit, that are not mutually exclusive and are the driving forces for human endeavor. Islam's holistic view of the human, i.e., integrating religion, spirituality and psychology, allows for spiritual caregivers to see their patients as one 'whole' and consider opening avenues of sacredness for them in the capacity that befits the patients. Hence, a characterization along the lines of *viewing the human as a whole and opening a window to the sacred*—which in the Muslim case ultimately is God—can be a starting point for a continued discussion in exploring a viable definition and scope of practice for Islamic spiritual care. This characterization is by no means full and cannot grasp the richness of the

terms addressed in the article. Considering the human being as body and spirit, material and spiritual, reveals humankind's holistic nature; therefore, the word 'whole' is in this description. Since care, in many ways (but not solely), is an action, a doing, 'opening a window' to spiritual, mental and emotional realities and experiences, it becomes part of the care provider's responsibility. 'Opening a window' is obviously metaphorically meant, implying that the care recipient may see new avenues of meaning and hope. The 'sacred' in this characterization can also be understood in the broadest of terms. The 'sacred' is whatever or whoever is closest to one's heart—the spiritual heart. Nevertheless, it is also important not to forget the richness and depth of spiritual expressions, not only outside of Islam but also within Islam, in order to avoid personal biases, stereotypes and spiritual impositions in spiritual care work.

**Author Contributions:** Methodology, N.B.; Validation, N.I.; Formal analysis, N.B.; Investigation, N.I.; Resources, N.B.; Writing—review & editing, N.B. and N.I.; Supervision, N.I.; Project administration, N.B. All authors have read and agreed to the published version of the manuscript.

**Funding:** This research received no external funding.

**Institutional Review Board Statement:** Not applicable.

**Informed Consent Statement:** Not applicable.

**Data Availability Statement:** Not applicable.

**Conflicts of Interest:** The authors declare no conflict of interest.

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
