# Peer review of "Exploring Islamic Spiritual Care: What Is in a Name?"

_religions, doi:10.3390/rel14101256_

Round 1
Reviewer 1 Report
Thank you for an engaging and important article.
p2 l59 you use the term humanities, it would be more usual to call some of the things you mention such as psychology, social sciences.
Line 110, you need to cite some research if you make a statement like this.
L326 the Scottish model of generic spiritual care as the default for chaplaincy may also be useful to mention.
I think the article would benefit from a more explicit conclusion answering the question you say the article covers: how can Islamic spiritual care be understood today, and what use may it have for care providers of other faiths or no faiths?
The current conclusion is a little oblique and could be written in a more practical manner for chaplains or spiritual care practitioners to apply to their practice. It is more of an issue of signposting and reframing a little rather than significant additions.
I prefer the question you are asking as a title rather than what you have chosen but that is a person preference when screening articles for inclusion or exclusion. With your current title I would expect a more far reaching discussion of spiritual care.
Author Response
Thank you for an engaging and important article.
p2 l59 you use the term humanities, it would be more usual to call some of the things you mention such as psychology, social sciences.
We changed it to “social studies”.
Line 110, you need to cite some research if you make a statement like this.
The reference was provided.
L326 the Scottish model of generic spiritual care as the default for chaplaincy may also be useful to mention.
I think the article would benefit from a more explicit conclusion answering the question you say the article covers: how can Islamic spiritual care be understood today, and what use may it have for care providers of other faiths or no faiths?
The current conclusion is a little oblique and could be written in a more practical manner for chaplains or spiritual care practitioners to apply to their practice. It is more of an issue of signposting and reframing a little rather than significant additions.
I prefer the question you are asking as a title rather than what you have chosen but that is a person preference when screening articles for inclusion or exclusion. With your current title I would expect a more far reaching discussion of spiritual care.
Reviewer 2 Report
General
It is an interesting and relevant article. At some points I did have some trouble following exactly what point the authors want to make, and here and there the structure could be improved. See also the suggestions below. Generally, the article contains a lot of relevant information, but I would recommend putting more line in there and really working toward a point, the conclusion of which is a logical and insightful consequence.
When I finished the article I noticed that there is no mention of alternative medicine, different explanatory models of illness (is something a psychosis or does someone have a jinn). I am curious how Islamic spiritual caregivers deal with this. Is anything known about this?
Introduction
In the introduction the authors clearly state why their article is useful and possibly even necessary. However I miss a statement about whether there are other articles and studies in the field of Islamic spiritual care in mental health, or whether they are the first to describe this.
R28: “Although there is still a gap in the contemporary health care system to address spiritual aspects of health, i.e., the role of the spiritual heart (qalb) in mental health, spirituality is also becoming popular in health care.” I wonder what gap they refer to. There have been quite a lot studies concerning spiritual care in mental health, also concerning gaps in this area, but also concerning the effect of spiritual interventions. Do the authors refer to Islamic spiritual care – considering the footnote? I think it may be worthwhile not to cite other studies in a footnote but in the text.
R31: “This is due to the fact that health care practitioners are aware that without spirituality, an individual may suffer from spiritual distress that "occurs when the person does not find meaning, hope, love, peace, comfort, on connections in life, or when they experience a conflict between what they believe and what actually happens in their lives" (Levitt 2005, p.63).” Is it true that persons with spiritual distress are individuals without spirituality? I think this passage could be worded differently. A second question about this statement is whether health care practitioners indeed are generally aware of this. Is there scientific information about this? Or is this an area that needs more attention?
Methodology and method
R58: “In this article, we use and discuss voices from different fields of humanities, like sociology, theology, and psychology, to give a theoretical overview of our understanding of Islamic spiritual care.” This is an interesting and promising sentence, but I also wonder how this would be possible and what the authors exactly mean here. I also would like to know how the authors proceeded in collecting material and writing this article.
R68: “Which '''authority' ensures healing or curing at hospitals?” This is an interesting question, but I wonder what to be its meaning in the context of this paragraph. The authors discuss authorities like ideologies or moral values, which, in my view is different from a possible authority that is beyond our ideologies, beliefs. Do the authors refer to a Higher Power here? And what would be the relation with the authorities when ideologies are meant?
R71: “These and other pertinent questions are part of the ongoing and heated discussions where borders are being drawn and redrawn to accommodate the different 'authorities’ at public institutions like hospitals.” I cannot fully understand how the subsequent sentences are examples of ‘heated discussions’. Could the authors explain this?
R88: “As a hermeneutics of Islamic spiritual care”, in my experience, hermeneutics is the study of written texts, so I cannot completely understand what the authors mean to say here. Also the sentence following, about the correlations, it is my question to what extent the correlations mentioned could answer the research question.
R92: “We correlate our many years of spiritual care practice”, I thought I read earlier that one of the two (R45) authors was a psychotherapist (R56)? Or are there three authors?
Theology and psychology
R108: “However, the research finds that Christian chaplains use and practice tools of psychology to serve their own theology, not the other way around.” I wonder: is that problematic to the authors or not? Could the authors elaborate a bit on this?
Definition of spirituality
I agree that the definition of spirituality is a choice. I wonder whether the authors have consciously chosen to leave a definition of the term ‘religion’ out?
R142: “In Islamic tradition…” till “…meaningful in the present.” I think this paragraph can be shortened.
Spiritual reality- beyond religious formality
The authors raise an interesting point by discussing the difference in appreciation of religion and spirituality in psychology. As a reader I expect some connotations, or some discussion of the separation of both. However, a kind of a new explanation of ‘spirituality’ follows, whereas the authors had already defined spirituality. My suggestion would be to cluster definitions of spirituality in the same paragraph and at this point to discuss the (un)appropriateness of the separation of religion and spirituality.
Spirituality, meaning and purpose
R187: “Such an approach might disregard the role of the process of meaning-making during spiritual care, it is still helpful to relate spirituality to meaning and purpose as a dominant theme.” This sentence is unclear, can the authors clarify what they mean? The authors continue with conclusions regarding the "definition point. I would suggest moving such a definition to the text that reflects their own definition, and elaborating on the sense of 'meaning' in spirituality in this place. Again, the last two sentences seem to reflect some kind of definitions or characteristics of spirituality. This is a bit confusing for the reader. I would suggest moving the current paragraph to the definition paragraph, either by combining the two or clearly separating them based on their headings.
R216/217: “These are the heart (qalb), intellect (aql), soul or self (nafs) and 216 spirit (ruh)” The ruh is mentioned again, but I think that is not intended here (?). I think the distinction between nafs and ruh is relevant to explain.
Spirituality and Sufi cosmology
R264: “Based on the Qur’anic challenge, all Muslim scholars unanimously agree that human beings cannot comprehend certain realities regarding the spirit (Elahi 2007).” I try to understand this sentence and paragraph in relation to the foregoing text about love. Could the authors relate the subparagraphs to each other?
R301/302: “Islamic spiritual care is based on the concept that humans are a composite of integral physiological, psychological, mental, and spiritual components.” What is meant with the difference between psychological and mental?
Islamic Spiritual Care in contemporary times
In this paragraph the authors switch from mentioning several characteristics of Islamic spiritual care, to an ‘restricted view’ as I interpret the passage about the not interchangeable vision and they finish the paragraph with the question how Islamic spiritual care should take shape in western society. As a reader, I am trying to connect these parts and I think I am missing the line a bit - perhaps it could be clarifying if the various sub-paragraphs were linked together so that it is clear how they relate to each other.
R326: “Exploring the humanist chaplaincy function may provide fruitful insights to understand chaplaincy in secularized societies. This is because humanist chaplaincy does not fit within the more traditional, religion-focused views of the profession which has several decades of history in Holland and institutional presence in Northwestern Europe (Schuhmann et al 2021). I agree that humanist chaplaincy and perhaps the more interfaith spiritual care may be worthwhile to study, but I am curious what would be the goal in this context? The authors mention they want to understand Islamic spiritual care in secularized society, but would they like to learn from humanist chaplaincy? I also do not fully understand ‘this is because’ of the last sentence. Possibly the authors can clarify what they mean.
Islamic Spiritual Care and Its Relations to Other Health Care Professions
R 368: “In human history” – in Islamic history?
It is an interesting way of thinking to talk about spiritual caregivers as psychotherapists, but the point would raise lots of questions and needs explanation. Does this only apply to ISC, or would you characterize all spiritual caregivers this way? And what then is the added value of the (possibly Muslim) psychotherapist? I would like to see this point elaborated a bit more: what would be the consequences, what argues for it and what argues against it?
Islamic spiritual care and its importance for care providers from different faith traditions- universal or specialized spiritual care?
This heading is a bit long and it is a question in itself. I would recommend to reduce the words and not the have a question as heading.
R413: The authors suddenly use ‘ISC’ – possibly that can be done earlier and explained the first time. The authors also mention the term ‘faith’, whereas not al spiritual caregivers would assume to have a faith. Possibly the term ‘outlook on life’ or something like that can be used.
R419: “ISC can also be used for care recipients that are not Muslims.” How would the authors envision this?
R425-445 again seem to refer back to authoritative Islamic writings on which the ISC is based. I wonder if it is in the most convenient place here, it might fit better in an earlier piece that elaborates the "theory" more. It seems more logical to me if the recommendations follow after the theory.
Conclusion
The conclusion is generally worded and focuses on the holistic perspective, which has attention in more philosophical groups. My preference would be to be more specific and name concrete points where the ISC adds value to spiritual care in general.
Author Response
General
It is an interesting and relevant article. At some points I did have some trouble following exactly what point the authors want to make, and here and there the structure could be improved. See also the suggestions below. Generally, the article contains a lot of relevant information, but I would recommend putting more line in there and really working toward a point, the conclusion of which is a logical and insightful consequence.
When I finished the article I noticed that there is no mention of alternative medicine, different explanatory models of illness (is something a psychosis or does someone have a jinn). I am curious how Islamic spiritual caregivers deal with this. Is anything known about this?
We have addressed this and provided reference to recent research studies in the field of Islamic spiritual care and psychotherapy.
Introduction
In the introduction the authors clearly state why their article is useful and possibly even necessary. However I miss a statement about whether there are other articles and studies in the field of Islamic spiritual care in mental health, or whether they are the first to describe this.
We have added the most relevant research in Introduction.
R28: “Although there is still a gap in the contemporary health care system to address spiritual aspects of health, i.e., the role of the spiritual heart (qalb) in mental health, spirituality is also becoming popular in health care.” I wonder what gap they refer to. There have been quite a lot studies concerning spiritual care in mental health, also concerning gaps in this area, but also concerning the effect of spiritual interventions. Do the authors refer to Islamic spiritual care – considering the footnote? I think it may be worthwhile not to cite other studies in a footnote but in the text.
We have added more clarification on the gaps in the health care system.
R31: “This is due to the fact that health care practitioners are aware that without spirituality, an individual may suffer from spiritual distress that "occurs when the person does not find meaning, hope, love, peace, comfort, on connections in life, or when they experience a conflict between what they believe and what actually happens in their lives" (Levitt 2005, p.63).” Is it true that persons with spiritual distress are individuals without spirituality? I think this passage could be worded differently. A second question about this statement is whether health care practitioners indeed are generally aware of this. Is there scientific information about this? Or is this an area that needs more attention?
We have revised and reworded this sentence.
Methodology and method
R58: “In this article, we use and discuss voices from different fields of humanities, like sociology, theology, and psychology, to give a theoretical overview of our understanding of Islamic spiritual care.” This is an interesting and promising sentence, but I also wonder how this would be possible and what the authors exactly mean here. I also would like to know how the authors proceeded in collecting material and writing this article.
We reworded “humanities” with “social studies.” We also added this to R58 to clarify how we collected material or this article: “We gathered relevant texts using search engines and databases such as ATLA Religion Database with ATLASerials (EBSCOhost databases), PubMed, and PsycINFO.”
R68: “Which '''authority' ensures healing or curing at hospitals?” This is an interesting question, but I wonder what to be its meaning in the context of this paragraph. The authors discuss authorities like ideologies or moral values, which, in my view is different from a possible authority that is beyond our ideologies, beliefs. Do the authors refer to a Higher Power here? And what would be the relation with the authorities when ideologies are meant?
We revised this sentence to make it more explisit.
R71: “These and other pertinent questions are part of the ongoing and heated discussions where borders are being drawn and redrawn to accommodate the different 'authorities’ at public institutions like hospitals.” I cannot fully understand how the subsequent sentences are examples of ‘heated discussions’. Could the authors explain this?
We deleted “heated” to avoid misunderstanding in this sentence.
R88: “As a hermeneutics of Islamic spiritual care”, in my experience, hermeneutics is the study of written texts, so I cannot completely understand what the authors mean to say here. Also the sentence following, about the correlations, it is my question to what extent the correlations mentioned could answer the research question.
Recently, in practical theology there is a discussion that the way we study religious texts we also need to “read and” “interpret” our patients. This is especially the concept of Anton Boisen in 1920s.
R92: “We correlate our many years of spiritual care practice”, I thought I read earlier that one of the two (R45) authors was a psychotherapist (R56)? Or are there three authors?
One author who is a psychotherapist is also spiritual care provider.
Theology and psychology
R108: “However, the research finds that Christian chaplains use and practice tools of psychology to serve their own theology, not the other way around.” I wonder: is that problematic to the authors or not? Could the authors elaborate a bit on this?
We revised this.it is reworded as: “In the last few decades, psychologists also used spirituality and religion to understand existential meaning assigned to suffering (Pargament 2007).”
Definition of spirituality
I agree that the definition of spirituality is a choice. I wonder whether the authors have consciously chosen to leave a definition of the term ‘religion’ out?
We decided to use spirituality to be more inclusive.
R142: “In Islamic tradition…” till “…meaningful in the present.” I think this paragraph can be shortened.
We hope to keep the paragraph as it is to explain Asad’s discursive tradition more explicitly in Islamic spiritual care.
Spiritual reality- beyond religious formality
The authors raise an interesting point by discussing the difference in appreciation of religion and spirituality in psychology. As a reader I expect some connotations, or some discussion of the separation of both. However, a kind of a new explanation of ‘spirituality’ follows, whereas the authors had already defined spirituality. My suggestion would be to cluster definitions of spirituality in the same paragraph and at this point to discuss the (un)appropriateness of the separation of religion and spirituality.
We shortened the definition of spirituality and added some discussion regarding religion and psychology and their shared interest in human experience and needs.
Spirituality, meaning and purpose
R187: “Such an approach might disregard the role of the process of meaning-making during spiritual care, it is still helpful to relate spirituality to meaning and purpose as a dominant theme.” This sentence is unclear, can the authors clarify what they mean? The authors continue with conclusions regarding the "definition point. I would suggest moving such a definition to the text that reflects their own definition, and elaborating on the sense of 'meaning' in spirituality in this place. Again, the last two sentences seem to reflect some kind of definitions or characteristics of spirituality. This is a bit confusing for the reader. I would suggest moving the current paragraph to the definition paragraph, either by combining the two or clearly separating them based on their headings.
We have moved the definitions to the section in the text where we discuss definition of spirituality. In this section, we have clarified the role of spirituality in creating meaning and purpose in significant life events and daily problems.
R216/217: “These are the heart (qalb), intellect (aql), soul or self (nafs) and 216 spirit (ruh)” The ruh is mentioned again, but I think that is not intended here (?). I think the distinction between nafs and ruh is relevant to explain.
We reworded it to bring more clarity to this sentence.
Spirituality and Sufi cosmology
R264: “Based on the Qur’anic challenge, all Muslim scholars unanimously agree that human beings cannot comprehend certain realities regarding the spirit (Elahi 2007).” I try to understand this sentence and paragraph in relation to the foregoing text about love. Could the authors relate the subparagraphs to each other?
We have restructed this paragrahh for more clarification.
R301/302: “Islamic spiritual care is based on the concept that humans are a composite of integral physiological, psychological, mental, and spiritual components.” What is meant with the difference between psychological and mental?
Psychological component is about emotions, fear, etc., whereas mental is about cognitions, cognitive abilities, etc.
Islamic Spiritual Care in contemporary times
In this paragraph the authors switch from mentioning several characteristics of Islamic spiritual care, to an ‘restricted view’ as I interpret the passage about the not interchangeable vision and they finish the paragraph with the question how Islamic spiritual care should take shape in western society. As a reader, I am trying to connect these parts and I think I am missing the line a bit - perhaps it could be clarifying if the various sub-paragraphs were linked together so that it is clear how they relate to each other.
R326: “Exploring the humanist chaplaincy function may provide fruitful insights to understand chaplaincy in secularized societies. This is because humanist chaplaincy does not fit within the more traditional, religion-focused views of the profession which has several decades of history in Holland and institutional presence in Northwestern Europe (Schuhmann et al 2021). I agree that humanist chaplaincy and perhaps the more interfaith spiritual care may be worthwhile to study, but I am curious what would be the goal in this context? The authors mention they want to understand Islamic spiritual care in secularized society, but would they like to learn from humanist chaplaincy? I also do not fully understand ‘this is because’ of the last sentence. Possibly the authors can clarify what they mean.
Islamic Spiritual Care and Its Relations to Other Health Care Professions
R 368: “In human history” – in Islamic history?
We revised it to “Islamic tradition”.
It is an interesting way of thinking to talk about spiritual caregivers as psychotherapists, but the point would raise lots of questions and needs explanation. Does this only apply to ISC, or would you characterize all spiritual caregivers this way? And what then is the added value of the (possibly Muslim) psychotherapist? I would like to see this point elaborated a bit more: what would be the consequences, what argues for it and what argues against it?
We have revised the paragraph.
Islamic spiritual care and its importance for care providers from different faith traditions- universal or specialized spiritual care?
This heading is a bit long and it is a question in itself. I would recommend to reduce the words and not the have a question as heading.
R413: The authors suddenly use ‘ISC’ – possibly that can be done earlier and explained the first time. The authors also mention the term ‘faith’, whereas not al spiritual caregivers would assume to have a faith. Possibly the term ‘outlook on life’ or something like that can be used.
We changed it to Islamic spiritual care.
We also revised “faith” to “spiritual or religious beliefs”.
R419: “ISC can also be used for care recipients that are not Muslims.” How would the authors envision this?
We had a brief statement to bring more clarity to this statement.
R425-445 again seem to refer back to authoritative Islamic writings on which the ISC is based. I wonder if it is in the most convenient place here, it might fit better in an earlier piece that elaborates the "theory" more. It seems more logical to me if the recommendations follow after the theory.
We have revised this paragraph adding names of the contemporary Muslim scholars.
Conclusion
The conclusion is generally worded and focuses on the holistic perspective, which has attention in more philosophical groups. My preference would be to be more specific and name concrete points where the ISC adds value to spiritual care in general.
We have added more nuances to the Conclusion section.